# Catchment Soil Properties Affect Metal(loid) Enrichment in Reservoir Sediments of German Low Mountain Regions



Jens Hahn [1,*], Thanh Bui [2], Mathias Kessler [2], Collin J. Weber [3], Thomas Beier [3], Antje Mildenberger [2], Martina Traub [1] and Christian Opp [3]

1    Federal Institute of Hydrology, Am Mainzer Tor 1, 56068 Koblenz, Germany; traub@bafg.de
2    Department of Geography, Institute for Integrated Natural Sciences, University of Koblenz-Landau, Universitätsstr. 1, 56070 Koblenz, Germany; thanh.bui2408@gmail.com (T.B.); mathias-.kessler@web.de (M.K.); mildenberger@uni-koblenz.de (A.M.)
3    Faculty of Geography, Philipps-Universität Marburg, Deutschhausstr. 10, 35037 Marburg, Germany; collin.weber@geo.uni-marburg.de (C.J.W.); beier.thomas@yahoo.de (T.B.); opp@staff.uni-marburg.de (C.O.)
*    Correspondence: jens.hahn@bafg.de

**Abstract:** Sediment management is a fundamental part of reservoir operation, but it is often complicated by metal(loid) enrichment in sediments. Knowledge concerning the sources of potential contaminants is therefore of important significance. To address this issue, the concentrations and the mobile fractions of metal(loid)s were determined in the sediments and the respective catchment areas of six reservoirs. The results indicate that reservoirs generally have a high potential for contaminated sediment accumulation due to preferential deposition of fine particles. The median values of the element-specific enrichment factor (EF) demonstrates slight enrichments of arsenic (EF: 3.4), chromium (EF: 2.8), and vanadium (EF: 2.9) for reservoir sediments. The enrichments of cadmium (EF: 8.2), manganese (EF: 3.9), nickel (EF: 4.8), and zinc (EF: 5.0) are significantly higher. This is enabled by a diffuse element release from the soils into the impounded streams, which is particularly favored by soil acidity. Leaching from the catchment soils partially enriches elements in stream sediments before their fine-grained portions in particular are deposited as reservoir sediment. We assume that this effect is of high relevance especially for reservoirs impounding small streams with forested catchments and weakly acid buffering parent material of soil formation.

**Keywords:** reservoir; soil; catchment; sediment; metals; trace elements; enrichment; element mobility

## 1. Introduction

Reservoirs are increasingly used for energy and drinking water production worldwide. Only in Germany, are there 371 "larger" dams (defined by either a height $\geq$ 15 m or a reservoir storage volume $\geq$ 3,000,000 m$^3$), the majority of which were built between 1900 and 2000 [1]. These reservoirs are mainly used for energy production and drinking water supply, and they are operated by private companies or federal states [2,3]. In addition, several reservoirs are used to regulate the water level of rivers and thus represent important instruments for the operation of federal waterways [4]. Dam rehabilitation, sediment dredging from the reservoir bottom, and other measures that serve to maintain their functionality are necessary works [5] that have to be increasingly carried out in recent years since most of the reservoirs in Germany are older than 80–100 years [2].

European regulations consider sediments as waste when they are removed from a waterbody [6]. Thus, dredged reservoir sediments have to be recycled or deposited in landfills. Due to continuous sedimentation and the increasing ages of dams, the utilization and disposal of reservoir sediments is a task of increasing importance beyond Germany, and beyond Europe [7–9]. Reservoir sediments characteristically act as a sink for contaminants [10]. This phenomenon can be explained by the fact that fine-grained, organic-rich sediments are capable of binding large amounts of potential contaminants

such as metal(loid)s and reservoirs are the preferred deposition areas of such fine sediments within running water systems [11,12]. Due to the high enrichment potential of metal(loid)s, dredged material from reservoirs often exceeds legal limits that must be met for recycling or landfilling [13,14]. In Germany, for example, this concerns the precautionary values of the Federal Soil Protection Ordinance, which regulate the application to arable land [15] (Table S1).

The enrichment of elements in reservoir sediments is mostly attributed to (historical) anthropogenic emissions that may result, for example, from atmospheric deposition, geogenic enrichments, historical mining, sewage water inlets, or the influence of further local point sources [16–20]. The long-term and continuous release of elements from catchment soils and their subsequent deposition in impoundments is usually less frequently considered as a factor of element enrichment in reservoir sediments. However, the fact that elevated concentrations of trace elements can also occur in reservoirs where an affection by geogenic enrichments, anthropogenic point sources, or atmospheric depositions is supposed to be very low [e.g., drinking water reservoirs] leads to the assumption, that significant amounts of metal(loid)s may be continuously transferred from catchments into reservoirs. The properties of catchment soils are expected to become increasingly important as a determinant of reservoir sediment quality, particularly in regions of retrogressive anthropogenic metal(loid) emissions. In order to prevent future management problems of the dredged material, it is crucial to find the connections between catchment soils and reservoir sediment quality.

In this study, the fate of metal(loid)s in forest dominated catchments of German low mountain ranges was investigated. The aims of the study were to describe (1) to which concentrations metal(loid)s are present in representative reservoirs of German low mountain regions, (2) which elements are preferentially enriched in reservoir sediments, and (3) which soil and sediment properties decisively influence the element transfer originating from catchment soils.

## 2. Materials and Methods

### 2.1. Regional Setting

Due to limited groundwater availability, the majority of German reservoirs are located in low mountain regions in the federal states of North Rhine-Westphalia (NRW) and Saxony (SN) [2]. The low mountain regions of both federal states are mainly part of the European Variscides. In NRW, the Variscides are represented by the geologic unit of the Renohercynian Zone, whereas in Saxony they are represented by the Saxothuringian Zone [21,22] (Figure 1). In order to achieve a high representativity, the selection of the study areas was focused on both these geological units, where most of the German reservoirs are located.

All six investigated reservoirs are located in low mountain ranges, which include areas of the Rhenish Slate Mountains (Bitburg Reservoir, Steinbach Reservoir, Breitenbach Reservoir, Obernau Reservoir) and of the Ore Mountains (Einsiedel Reservoir, Lehnmühle Reservoir). The sites of the Rhenish Slate Mountains are characterized by sub-oceanic climatic conditions and a dominance of Late to Middle Devonian rocks. The sites of the Ore Mountains reservoirs are characterized by the gradual transition to a more continental climate and late Proterozoic rocks (Table 1). The Bitburg, Breitenbach, and Obernau reservoirs are characterized by a geology largely comprising altering layers of sand, silt, claystone, greywacke, and slate, whereby limestone is restricted to a few small parts of the Bitburg Reservoir catchment. Although also situated within the Rhenish Slate Mountains, the catchment of the Steinbach Reservoir comprises mainly quartzite and slate as geologic basement [23]. Both reservoirs situated within the Ore Mountains are characterized by a dominance of metamorphic rocks, which mainly comprise phyllite (Einsiedel Reservoir) and gneiss (Lehnmühle Reservoir) [24,25]. Due to a widespread occurrence of silicate rocks, forestry, and a humid climate, the catchments of all reservoirs are characterized by soilscapes dominated by Cambisols and Stagnosols [26–28].

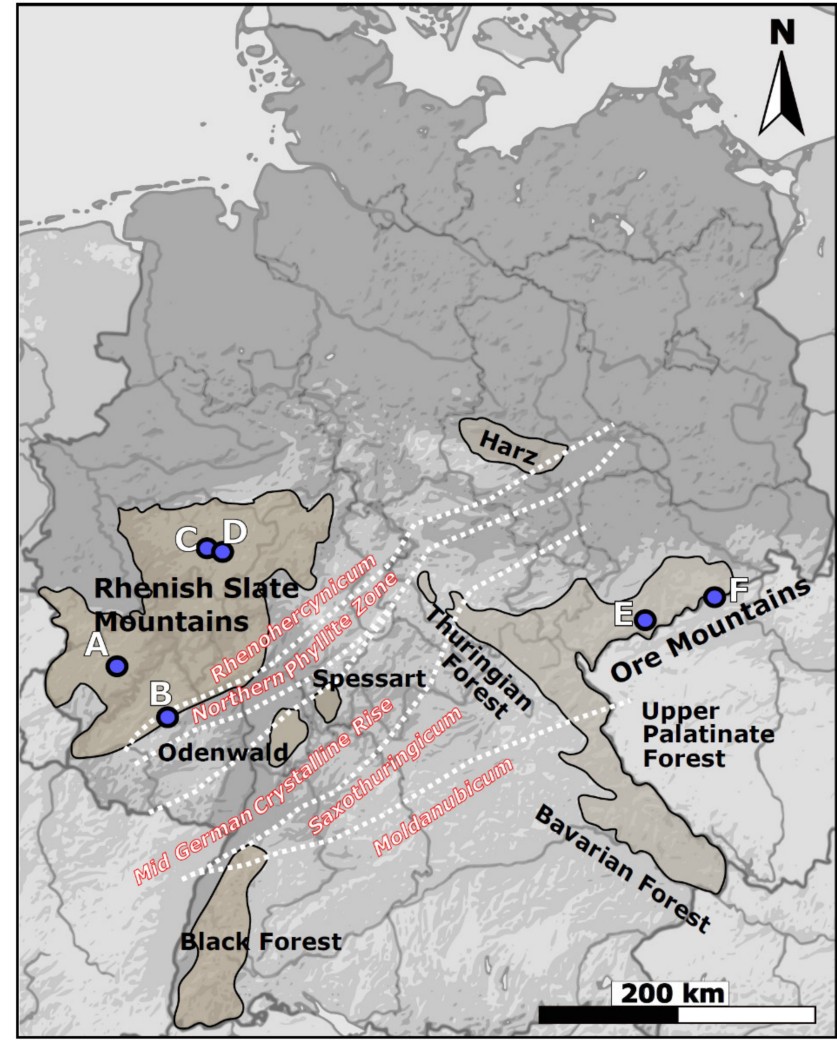

**Figure 1.** Study areas within the geologic units of the German Variscides. A: Bitburg Reservoir; B: Steinbach Reservoir; C: Breitenbach Reservoir; D: Obernau Reservoir; E: Einsiedel Reservoir; F: Lehnmühle Reservoir.

**Table 1.** Properties of the investigated reservoirs and their respective catchments (Constr.: period of construction; Vol.: Storage volume; Prec.: mean annual precipitation; Temp.: mean annual temperature; Elev.: elevation above sea level; Size: size of the respective catchment of a reservoir).

| Reservoir | Einsiedel | Lehnmühle | Steinbach | Bitburg | Breitenbach | Obernau |
|---|---|---|---|---|---|---|
| **Constr. (years)** | 1891–1894 | 1927–1931 | 1963–1966 | 1970–1972 | 1953–1956 | 1967–1972 |
| **Vol. (Mill. m³)** | 0.3 | 21.9 | 4.5 | 1.2 | 7.8 | 14.8 |
| **Prec. (mm)** | 701 [a] | 664 [b] | 517 [c] | 861 [d] | 783 [e] | 783 [e] |
| **Temp. (°C)** | 8.5 [a] | 9.4 [b] | 9.5 [c] | 9.0 [d] | 9.2 [e] | 9.2 [e] |
| **Elev. (m a.s.l.)** | 384 | 518 | 465 | 256 | 370 | 370 |
| **Geology** | phyllite | gneiss | shale, quartzite | shale, sand-, limest. | shale, sandst. | shale, sandst. |
| **Size (km²)** | 2.7 | 60.4 | 14.6 | 330 | 4.1 | 11.3 |

Monitoring stations: [a] Chemnitz (WeWa) (1980–2010) [28]; [b] Dresden Klotz: (1980–2010) [29]; [c] Bad Kreuznach (1961–1990) [30]; [d] Olsdorf (1980–2010) [29]; [e] Siegen Kläranlage (1980–2010) [29].

The reservoirs are mainly used for drinking water supply. The only exception is the Bitburg Reservoir that was, although used for water regulation and energy production, included in the study because of mainly similar catchment characteristics. Although the study objects clearly differ in age and storage volume, all reservoirs are characterized

by catchment areas with forestry being the dominant land use and the impounding of relatively small watercourses. In case of the Einsiedel, Obernau, and Breitenbach reservoirs additional water supplies are provided by intake tunnels, whose impacts on sediment quality can, however, be omitted.

### 2.2. Soil and Sediment Sampling

The theoretical background of the study is based on research by Yang et al. (2016), which demonstrates that inputs originating from a catchment can be an important source of contaminants to lakes [31]. To enable an assessment of the metal(loid) transfer between catchments and reservoirs, the sampling strategy was to compare the element contents of reservoir sediments (RS), of sediments of the impounded streams (SS), and of the catchment soils (CS). This allows identification of enrichment tendencies within a respective sample group and the possible links between the potential element release by CS and element enrichments in SS and/or RS [32]. However, due to differences in catchment size, the number or morphology of influents, or the accessibility of potential sampling sites, sampling had to be adapted to each study region, which resulted in different sample numbers between the study objects. Since the total number of each sample group is relatively high, regional differences in sample numbers were omitted.

Samples of RS were collected during 2018 (Einsiedel Reservoir) and 2020 (other reservoirs) from each main basin of a dam as disturbed samples from 0–15 cm depth. This was done in the Bitburg, Steinbach, Obernau, and Breitenbach reservoirs using a Van Veen sampler. Samples from the Einsiedel and Lehnmühle reservoirs were taken by a spade shortly after the water level was reduced during rehabilitation work.

Samples of SS were collected in flow-calmed areas from 0–15 cm depth as disturbed samples using a shovel. When possible, samples were collected from each stream contributing to a reservoir's stored water. The number of samples, therefore, varied depending on the formation of the water networks which were located upstream to the reservoirs. At some locations, the sampling was partly restricted by the occasional absence of fine sediment, which was in particular the case in inlets enclosed by amourstones or concrete slabs.

Soil investigation was based on sites located in close proximity (0–5 m) to the impounded streams or their tributaries, where an element transfer by erosion or interflow was assumed to be possible. Sampling was restricted on terrestrial valley slopes (=terrestrial soils) and the fluvial impacted valley bottoms (semi-terrestrial soils) of forested sites. The sampling was additionally focused on soil types which were found to be representative for a respective catchment. All studied soils were classified according to the IUSS Working Group WRB (2015) classification in the field [33]. Disturbed soil samples were collected at depths of 0–10 cm, 20–30 cm, and 50–60 cm depth using an Edelmann auger. All samples were transported in polyethylene bags and air-dried. With the exception of the Einsiedel and Lehnmühle reservoirs, aliquots of RS samples were additionally frozen at $-18\ ^\circ$C until analysis.

### 2.3. Sample Processing

All samples were homogenized in a porcelain mortar and sieved (<2 mm) before analyses. To analyze the total organic carbon (TOC) and the total nitrogen ($N_t$), the samples were dried for 16 h at 105 $^\circ$C and ground in a pebble mill. The concentrations of TOC and $N_t$ were determined by a C/N analyzer (vario EL cube, Elementar, Langenselbold, Germany) after combusting the samples at 950 $^\circ$C [34].

Grain size analysis was conducted by suspending 0.3–1 g of a respective sample in water added with 1 mL of a dispersing agent (4% $Na_4P_2O_7$). Each suspension was sonicated for 10 min. Grain size distributions were subsequently determined by laser diffractometry (LA 960V2, Horiba). Acidity was determined by repeated shaking of the dried samples in 0.01 molar $CaCl_2$. After a reaction time of two hours, the pH value was measured by means of a glass electrode (WTW Weilheim, Germany) [35].

Metals and metalloids were extracted as pseudo-total concentrations by heating (85–90 °C, 2 h) 1 g of a respective sample in aqua regia from 15 mL of HCl (37%) and 5 mL $HNO_3$ (65%) [36]. The extracts were subsequently filtered with cellulose filters (particle retention 1–2 μm, Sartorius) and diluted with ultrapure water. Since a potential element release in soils can be estimated by their mobilizable element proportions, the easily exchangeable element concentrations were determined in ammonium-nitrate extractions. For this purpose, 10 g of each sample was shaken (1.5 h) in 25 mL of 1 M ammonium-nitrate [37]. Since this extraction is usually performed on dried samples and oxic conditions were found in the field for the vast majority of soils, it was performed on dried soil samples (CS). In contrast, the organoleptic characteristics of RS indicated highly anoxic conditions. To minimize the possible influences of drying on easily exchangeable element concentrations, these were determined on field-fresh (previously frozen) RS. The extracts were centrifuged at 3000 g, filtered, diluted to 50 mL with ultrapure water, and stabilized by the addition of 0.5 mL $HNO_3$ (65%). All extracts were stored in pre-cleaned polypropylene centrifuge tubes at 4 °C until analysis. The aqua regia extractable concentrations of aluminum (Al), chromium (Cr), iron (Fe), manganese (Mn), nickel (Ni), lead (Pb), and zinc (Zn) were analyzed by ICP-OES (Optima 8300, Perkin Elmer) using matrix adaption and a recovery control by certified reference material (NIST 1646a, ERM-CC020) [38]. As well as the ammonium-nitrate extractable concentrations of all elements, the concentrations of aqua regia extractable arsenic (As), cadmium (Cd), and vanadium (V) were determined in diluted extracts by ICP-QQQ-MS (8800, Agilent) [39].

The results of ICP-OES and ICP-QQQ-MS analyses were used to calculate the enrichment factor (EF), which relates the concentration of a metal(loid) to the concentration of a reference element of low variability and is suitable for assessing anthropogenic element enrichment [40]. Because its calculation normalizes sample-specific variations in grain sizes, it is useful for comparing different samples or groups of samples. Due to its comparatively low mobility under varying redox conditions, Al was used as a reference element. The EF was calculated according to Pourabadehei and Mulligan (2016) [41] by the equation:

$$EF = (M/Al)s/(M/Al)r$$

where (M/Al)s is the mean metal(loid)/Al ratio of a respective sample divided by (M/Al)r, which is the mean element-specific ratio to Al of European river sediments according to Salminen et al. (2005) [42], using the following values: Al = 5.4 (%); As = 9.5 (mg kg$^{-1}$); Cd = 0.527; Cr = 31 (mg kg$^{-1}$) Mn = 716 (mg kg$^{-1}$); Ni = 28.6 (mg kg$^{-1}$); Pb = 29.8 (mg kg$^{-1}$); V = 33 (mg kg$^{-1}$); Zn = 98 (mg kg$^{-1}$).

### 2.4. Statistics

The data collected were analyzed for existing correlations using IBM SPSS Statistics 23. All data were tested for normal distribution using the Kolmogorov–Smirnov Test. For the majority, no normal distribution was observed so correlation analysis was performed using Spearman's non-parametric rank correlation coefficient ($r_{sp}$). The correlation analysis included a 2-tailed significance test related to the significance levels $p \leq 0.05$ and $p \leq 0.01$. The differences in element concentration between the sample groups (CS, SS, RS) were analyzed for significance using the Kruskal–Wallis H test followed by a Dunn–Bonferroni post-hoc test. In order to identify similar functionalities of the analyzed parameters, a principle component analysis (PCA) was performed. Within the frame of this analysis, the investigated parameters were summarized into three different components. Kaiser–Meyer–Olkin and Bartlett's sphericity tests were applied to control the suitability of the dataset for a PCA. After the suitability was verified, commonalities of element-specific behavior were analyzed by PCA using an orthogonal varimax rotation.

## 3. Results

### 3.1. Sample Characterization

The investigated reservoir sediments (RS) and stream sediments (SS) have a mostly silty texture and variously pronounced organoleptic characteristics of anoxic conditions. This also applies to the sediments of the Einsiedel and Lehnmühle reservoirs, which were drained shortly before sampling. The investigated catchment soils (CS) of terrestrial slope areas were predominantly classified as Dystric Cambisols. Sporadically, Cambic Leptosols were also encountered. In the semi-terrestrial areas of the valley bottoms, the soils were classified as different variants of Fluvic Gleysols and Gleyic or Eutric Fluvisols. As a special feature, in the catchment area of the Steinbach Reservoir, the soil type Dystric Histosol was also found in some cases at semi-terrestrial sites.

The grain size of CS corresponds mostly to sandy silt. On average, SS are slightly coarser grained, being mostly classified as silty sand, while RS are characterized by the finest texture and mostly correspond to pure silt (Table 2).

**Table 2.** Mean value, median value, and standard deviation (SD) of: total organic carbon (TOC, in %), total nitrogen ($N_t$, in %), C/N-ratio, pH value, the volumetric percentage of sand (S), silt (U), and clay (C), and the pseudo-total element concentrations (Al and Fe in g kg$^{-1}$, other elements in mg kg$^{-1}$) of catchment soils (CS; $n$ = 89), stream sediments (SS; $n$ = 35), and reservoir sediments (RS; $n$ = 50).

|  | TOC | $N_t$ | C/N | pH | S | U | C | Al | As | Cd | Cr | Fe | Mn | Ni | Pb | V | Zn |
|---|---|---|---|---|---|---|---|---|---|---|---|---|---|---|---|---|---|
| **RS** | | | | | | | | | | | | | | | | | |
| Mean | 4.44 | 0.44 | 10.2 | 5.89 | 18.9 | 73.2 | 4.83 | 36.3 | 46.1 | 4.03 | 41.8 | 38.2 | 7764 | 78.6 | 121 | 46.8 | 343 |
| Median | 4.37 | 0.43 | 9.95 | 5.98 | 17.8 | 76.2 | 4.89 | 25.0 | 15.0 | 1.84 | 39.1 | 38.1 | 1171 | 81.2 | 88.3 | 42.6 | 299 |
| SD | 1.11 | 0.11 | 1.67 | 0.66 | 10.3 | 15.9 | 2.25 | 20.1 | 67.4 | 5.64 | 12.8 | 17.1 | 12,435 | 33.9 | 70.4 | 16.6 | 194 |
| **SS** | | | | | | | | | | | | | | | | | |
| Mean | 4.07 | 0.25 | 15.6 | 5.64 | 58.0 | 35.8 | 2.62 | 16.7 | 11.9 | 1.61 | 27.1 | 28.3 | 1654 | 69.2 | 90.6 | 26.5 | 131 |
| Median | 3.33 | 0.22 | 14.8 | 5.55 | 63.1 | 32.7 | 1.61 | 13.6 | 8.41 | 0.46 | 25.8 | 28.0 | 1238 | 59.9 | 40.8 | 22.9 | 107 |
| SD | 3.53 | 0.17 | 5.34 | 0.61 | 18.7 | 15.9 | 2.20 | 13.3 | 13.3 | 4.21 | 15.4 | 11.7 | 1813 | 62.1 | 279 | 19.3 | 107 |
| **CS** | | | | | | | | | | | | | | | | | |
| Mean | 5.84 | 0.35 | 14.8 | 3.84 | 33.1 | 58.7 | 6.54 | 19.2 | 15.0 | 0.43 | 27.6 | 25.7 | 756 | 30.6 | 87.9 | 29.5 | 76.0 |
| Median | 3.91 | 0.26 | 14.2 | 3.78 | 34.2 | 60.3 | 4.56 | 13.3 | 10.2 | 0.27 | 24.0 | 28.6 | 704 | 29.9 | 52.8 | 25.2 | 73.3 |
| SD | 6.37 | 0.28 | 3.68 | 0.59 | 18.2 | 15.5 | 5.27 | 17.9 | 23.1 | 0.51 | 17.4 | 11.6 | 553 | 15.3 | 84.9 | 22.5 | 51.9 |

The concentrations of TOC do not reveal any overarching patterns, and they are highly variable within all sample groups. The obviously high variability of TOC in CS is connected with the heterogeneity of the studied soil types. Maximal values of TOC are present within the Gleysols and Histosols of the Steinbach Reservoir catchment. The concentrations of $N_t$ are on average highest within RS, while $N_t$ in RS has a lower variability than in SS or CS. The C/N ratio is consistently lowest in mean values and variability within RS. It reaches significantly higher mean values and variability in SS and CS, without any apparent regularity.

In a comparison among the sample groups, in almost all study areas the highest pH values occur in RS. Usually, a slightly lower pH occurs in SS, while CS show significantly stronger acidic conditions.

In the case of all elements investigated, the element concentrations are averagely highest in RS, which illustrates the sink function of the reservoirs. The mean and median values of the elements Cd, Mn, Ni, and Zn increase in the order CS < SS, RS (Figure 2). In contrast, the elements Al, As, Cr, Fe, Pb, and V reach similarly high concentrations in SS and CS. In RS, they reach mostly higher concentrations and variations. All element concentrations are marked by significant differences ($p < 0.01$) between RS and SS or CS (Table 3).

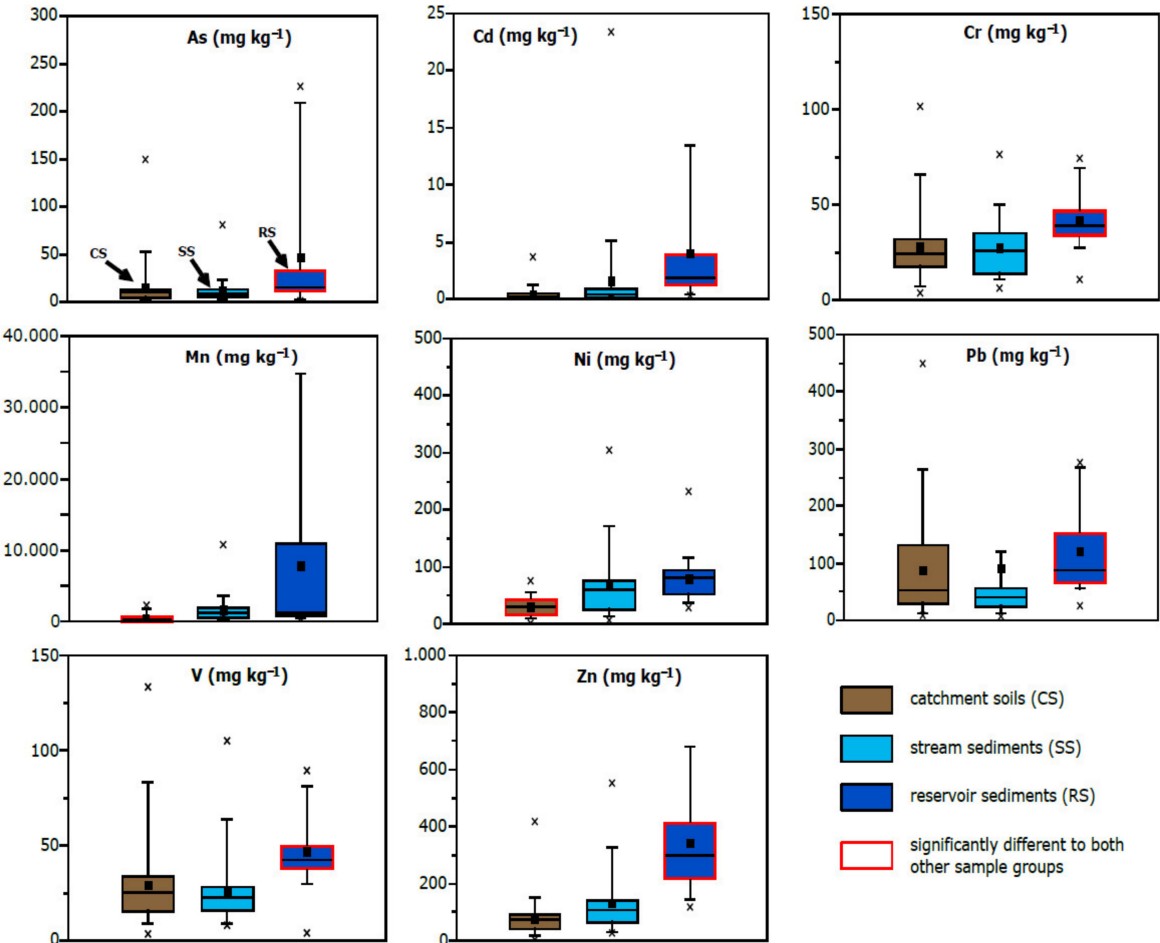

**Figure 2.** The element concentrations (As, Cd, Cr, Mn, Ni, Pb, V, Zn) of samples from catchment soils (CS, *n*: 89), river sediments (SS, *n*: 35), and reservoir sediments (RS, *n*: 50). The boxplots contain the element-specific concentrations of all sampling areas (black squares = mean values; crossbars = median values; x = minimum/maximum value; whiskers include the 5th to 95th percentile, the boxes include the 25th to 75th percentile). Significant differences were determined by Kruskal–Wallis H-test and Dunn–Bonferroni post-hoc test and are related to $p \leq 0.01$.

**Table 3.** Asymptotic significances (Kruskal–Wallis H test; $p \leq 0.01$ in bold letters) between the sample groups (CS, SS, RS) in reference on the element concentrations (Conc.) and the enrichment factors (EF).

| Asympt. Sig | Al | As | Cd | Cr | Fe | Mn | Ni | Pb | V | Zn |
|---|---|---|---|---|---|---|---|---|---|---|
| **Conc.** | **≤0.01** | **≤0.01** | **≤0.01** | **≤0.01** | **≤0.01** | **≤0.01** | **≤0.01** | **≤0.01** | **≤0.01** | **≤0.01** |
| **EF** | - | 0.74 | **≤0.01** | **≤0.01** | - | **≤0.01** | **≤0.01** | 0.13 | 0.29 | **≤0.01** |

### 3.2. Trace Metal Enrichment

In contrast to the pseudo-total concentrations, the element enrichment indicated by the enrichment factor (EF) partially demonstrates fewer differences between the different sample groups (Figure 3). In particular, the EF of Cr and V indicates rather minor enrichments in RS. In comparison to the different sample groups, the EF of V shows similar median values (median value: CS = 2.6; SS = 2.6; RS = 2.9) and variations without being significantly different to each other. For Cr, the enrichment in RS is significantly lower than in SS or CS (median value: CS = 3.0; SS = 3.2; RS = 2.8). The EF median values of As show

a similarly homogeneous picture in the comparison of the sample groups, whereby the enrichment is generally slightly higher (median value: CS = 3.7; SS = 3.8; RS = 3.4). The elements Cd, Ni, Pb, and Zn are enriched in a higher degree in SS and/or RS. According to the pattern of their pseudo-total concentrations, the EF of Cd (median value: CS = 1.9; SS = 2.7; RS = 8.2) and Zn (median value: CS = 2.4; SS = 3.6; RS = 5.0) indicate the lowest enrichments in the group of the CS, while RS are the most enriched. The elements Mn and Ni are rather slightly enriched in CS, reach significantly higher enrichments in RS, and they are the most enriched (Mn median value: CS = 3.1; SS = 7.2; RS = 3.9; Ni median value: CS = 3.6; SS = 6.5; RS = 4.8) in SS. Pb shows the highest median value (CS = 5.9; SS = 4.1; RS = 5.4) and the largest variation in the group of CS, although mean values and variation are less differently pronounced than for Cd, Ni, or Zn. In comparison to the pseudo-total concentrations, the presence of significant differences of the EF between the investigated sample groups is restricted to the elements Cr, Cd, Mn, Ni, and Zn (Table 3). In the case of Cr, significant differences rely on the significantly lower enrichment in RS. In contrast, significant differences of Cd, Mn, Ni, and Zn rely on the fact that the EF of SS and/or RS is significantly higher than the EF of CS.

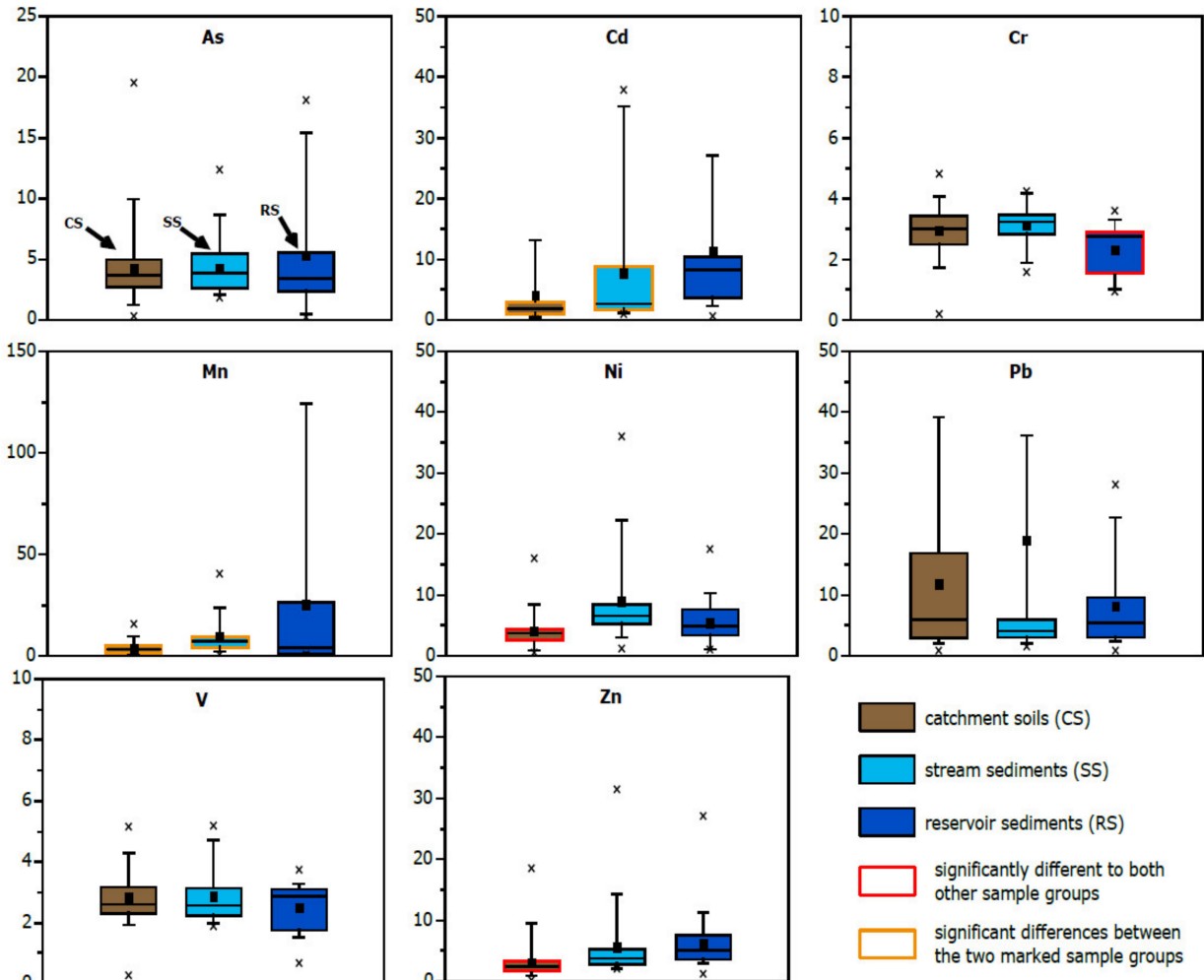

**Figure 3.** The element-specific enrichment factors (Al, As, Cd, Cr, Mn, Ni, Pb, V, Zn) of samples from catchment soils (CS, *n*: 89), river sediments (SS, *n*: 35), and reservoir sediments (RS, *n*: 50). The boxplots contain the element-specific enrichment factors of all sampling areas of a sample group (black squares = mean values; crossbars = median values; x = minimum/maximum value; whiskers include the 5th to 95th percentile, the boxes include the 25th to 75th percentile). Significant differences were determined by Kruskal–Wallis H-test and Dunn–Bonferroni post hoc test and are related to $p \le 0.01$.

The concentrations of TOC are only significantly correlated with the concentrations and EF of the elements Pb and Cd, being characterized by rather weak to moderate rank correlation coefficients ($r_{sp} < 0.5$) (Table 4). Additionally, rather weak correlations occur between $N_t$ and the concentrations as well as the EF of Cd, Pb, and Zn, whereby the correlations between $N_t$ and Cd reach the comparatively highest $r_{sp}$. Both the concentrations and the EF of Mn, Ni, Zn, and Cd show weak to medium positive correlations with the pH of the samples. The concentration of Al basically correlates positively with the quantities of all investigated metal(loid)s. Particularly strong correlations occur between the concentrations of Al and Cr ($r_{sp}$: 0.91) as well as between Al and V ($r_{sp}$: 0.89). The concentration of Fe is generally positively correlated with the concentrations of all the elements studied, with the strongest correlations to the elements Cr, Ni, and V. Significant correlations between the concentration of Fe and the EF of the elements generally fall below the correlation coefficient of $\leq 0.3$ used to declare a weak correlation.

**Table 4.** The element-specific Spearman rank correlation coefficients ($r_{sp}$) referred on the concentrations of total organic carbon (TOC), aluminum (Al), pH value, total nitrogen ($N_t$), and iron (Fe). Bold letters indicate significant correlations with $r_{sp} \geq 0.3$ or $r_{sp} \leq 0.3$ (* $p \leq 0.05$; ** $p \leq 0.01$).

| | As | EF_As | Cd | EF_Cd | Cr | EF_Cr | Mn | EF_Mn | Ni | EF_Ni | Pb | EF_Pb | V | EF_V | Zn | EF_Zn |
|---|---|---|---|---|---|---|---|---|---|---|---|---|---|---|---|---|
| **TOC** | −0.02 | −0.01 | **0.34**\*\* | **0.42**\*\* | −0.00 | 0.03 | −0.14 | **−0.20**\* | −0.02 | −0.08 | **0.32**\*\* | **0.36**\*\* | 0.04 | 0.13 | 0.06 | 0.11 |
| **N<sub>t</sub>** | **0.24**\*\* | 0.07 | **0.54**\*\* | **0.51**\*\* | **0.23**\*\* | −0.06 | 0.11 | −0.01 | **0.24**\*\* | 0.07 | **0.49**\*\* | **0.37**\*\* | **0.29**\*\* | **0.21**\* | **0.35**\*\* | **0.31**\*\* |
| **pH** | 0.13 | −0.02 | **0.49**\*\* | **0.44**\*\* | **0.23**\*\* | −0.06 | **0.58**\*\* | **0.51**\*\* | **0.64**\*\* | **0.58**\*\* | 0.03 | −0.09 | **0.21**\*\* | 0.04 | **0.59**\*\* | **0.67**\*\* |
| **Al** | **0.74**\*\* | - | **0.46**\*\* | - | **0.91**\*\* | - | **0.36**\*\* | - | **0.52**\*\* | - | **0.37**\*\* | - | **0.89**\*\* | - | **0.64**\*\* | - |
| **Fe** | **0.63**\*\* | 0.04 | **0.39**\*\* | 0.04 | **0.74**\*\* | **−0.25**\*\* | **0.63**\*\* | **0.25**\*\* | **0.70**\*\* | **0.19**\* | **0.41**\*\* | −0.10 | **0.70**\*\* | **−0.16**\* | **0.62**\*\* | **0.19**\* |
| *n* | 140 | 139 | 140 | 139 | 140 | 139 | 140 | 139 | 140 | 139 | 140 | 139 | 140 | 139 | 140 | 139 |

The results of the PCA explain 75.8% of variations by three components (Figure 4). Component 1 explains to a high degree (78–94%) the concentrations of Cr, Al, V, Fe, and As, while component 2 preferentially summarizes the variations of Zn, Cd, pH value, Ni, Mn, and Pb (46.3–80.3%). The dynamics of TOC, $N_t$, Cd, Pb, and C/N are partially explained by component 3 (32.8–95.7%), with TOC and $N_t$ showing the strongest relationship within this component (for details see Table S2).

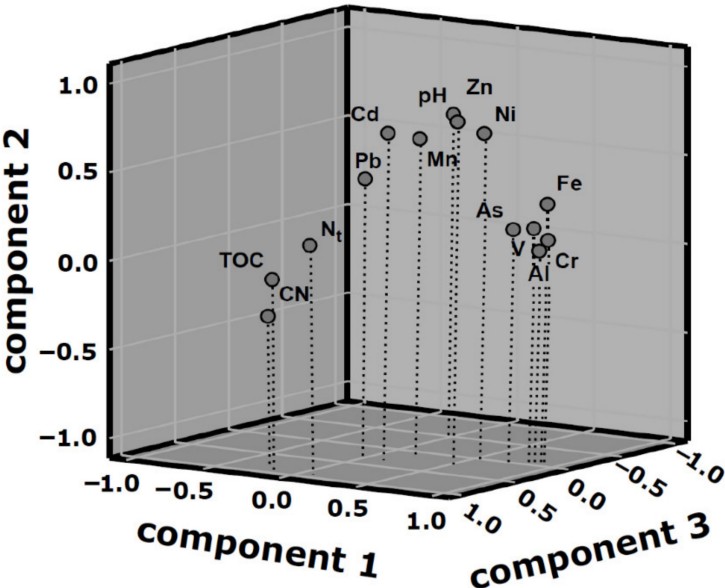

**Figure 4.** The component plot of the pseudo-total concentrations of trace elements (Al, As, Cd, Cr, Fe, Mn, Ni, Pb, V, Zn), total organic carbon (TOC), total nitrogen ($N_t$), C/N ratio (CN), and pH value in rotated space.

### 3.3. Trace Metal Mobility

The easily exchangeable percentages of the trace metal concentrations of the air-dried soil samples are relatively low, especially for As (<0.1–6.4%), Cr (<0.1–2.2%), and V (<0.1–1.6%) (Table 5). A slightly higher average percentage occurs for Ni (<0.1–57.1%), Pb (<0.1–51.6%), and Zn (<0.1–100%). Mean value, standard deviation, and maximal value indicate a higher soil-specific variation of Zn mobility in comparison to both Ni and Pb. The mobile percentage of Cd (<0.1–100%) and Mn (<0.1–100%) proves a comparatively high mobility for both elements. Contrarily, the field-fresh RS show very low (<1.0%) easily exchangeable proportions for most elements investigated. Although Mn (1.03–28.4%) reaches a lower mobility than recorded for CS, it proves to be the only element analyzed which keeps a relatively high mobility within the field fresh RS (for results of air-dried RS see Table S3).

**Table 5.** The percentage of easily extractable trace metal concentrations on the pseudo-total concentrations of air-dried samples of catchment soils (CS) and field fresh reservoir sediments (RS).

| | **CS (*Air–Dried*)** | | | | | | | | **RS (*Field Fresh*)** | | | | | | | |
|---|---|---|---|---|---|---|---|---|---|---|---|---|---|---|---|---|
| | **As** | **Cd** | **Cr** | **Mn** | **Ni** | **Pb** | **V** | **Zn** | **As** | **Cd** | **Cr** | **Mn** | **Ni** | **Pb** | **V** | **Zn** |
| **Mean (%)** | 0.76 | 48.1 | 0.33 | 42.5 | 11.2 | 5.97 | 0.17 | 20.1 | 0.64 | 0.04 | 0.01 | 13.1 | 0.48 | 0.04 | 0.01 | 0.11 |
| **Median (%)** | 0.36 | 47.1 | 0.25 | 32.7 | 5.75 | 3.95 | 0.06 | 11.0 | 0.32 | 0.02 | 0.01 | 14.1 | 0.33 | 0.02 | 0.01 | 0.08 |
| **SD** | 1.13 | 25.5 | 0.30 | 33.4 | 12.3 | 7.66 | 0.28 | 21.8 | 0.97 | 0.04 | 0.00 | 7.49 | 0.30 | 0.06 | 0.01 | 0.13 |
| *n* | 89 | 89 | 89 | 89 | 89 | 89 | 89 | 89 | 33 | 33 | 33 | 33 | 33 | 33 | 33 | 33 |

### 4. Discussion

The studied sample groups show some characteristic differences between their physico-chemical properties, from which the existing element enrichment differences can be partially explained. Differences in grain sizes already became apparent in the results of laser diffractometry. The volumetric proportions of the grain fractions prove the presence of the finest particles within the RS. The low clay concentrations of this sample group are attributed to the presence of organic fine particles, which are assumed to correspond predominantly to the silt fraction and are detected as mineral grains by laser diffraction. The fact that at least the RS must contain not insignificant amounts of clay is expressed in their high quantities of Al, whose concentration usually correlate strongly with the gravimetric proportion of clay minerals [43]. The concentrations of Al, which turn out to be much higher in the RS than in the CS or the SS, prove increasing concentrations of all studied elements associated with increased fine grain contents. This is evident in the numerous strong correlations, which are particularly pronounced for Al and the concentrations of Cr ($r_{sp}$: 0.91) and V ($r_{sp}$: 0.89). Together with As and Fe, similarities of Al, Cr, and V are also supported by the results of the PCA, which expressively summarizes these elements by the same component. The accumulation of trace elements in the fine fraction is a frequently described process [44–46], which explains the, generally, highest concentrations of all studied elements within the RS and proves the classical sink function of reservoirs. The fact that Cr and V are only slightly enriched within the study areas investigated agrees with the comparatively lowest mobility in the CS, which relies on a relatively strong binding in soils or sediments without relevant anthropogenic enrichments [47,48]. By largely matching geogenic background values for sedimentary and granitic rocks [49], the concentrations of both elements are interpreted to preferentially result from a geogenic origin and an almost exclusive deposition within the studied water bodies after erosion processes as described by Quinton and Catt (2007) [50]. Since Cr and V are rather immobile during anoxic conditions and neutral pH [51,52], the deposition of fine sediment is supposed to be the only relevant process controlling their enrichment in the studied reservoirs. A similar situation can be assumed for As, although it seems to be slightly lower depending on the accumulation of mineral fine particles, and it can be redistributed within the sediment under anoxic conditions [53]. The comparatively higher average EF is explained by higher concentrations

within reservoirs of the Ore Mountains, where an elevated geogenic background and historic anthropogenic activities [54,55] may have contributed to the enrichment of As at some locations in the study areas.

The lowest correlation coefficients of Al are found with the, comparatively, most mobile elements Cd ($r_{sp}$: 0.46) and Mn ($r_{sp}$: 0.36), as well as with Pb ($r_{sp}$: 0.37). The positive correlations of Pb with the concentrations of TOC and $N_t$ agree with the situation found for the Klingenberg Reservoir, Germany [32]. Since Pb has a relatively low element-specific mobility and no correlations with the pH value were found, it is interpreted that Pb is deposited in reservoirs in particular by the erosion of organic-rich topsoil material. Due to its ubiquitous enrichment resulting from atmospheric depositions [56], particularly in forest soils [57], it can be assumed that it is transported to the reservoirs in a similar way as Cr and V. However, due to a comparatively higher preference to be bound to soil organic matter [58,59], the deposition of Pb in reservoir sediments seems to be influenced to a higher degree by the sedimentation of organic detritus.

The enrichment of Cd, Mn, Ni, and Zn is, contrarily, not explainable by soil erosion and subsequent sedimentation within the aquatic system alone. The deposition of fine sediments in reservoirs generally favors the enrichment of these elements, which agrees with the frequent presence of their highest concentrations in RS. However, the EF, by which the individual samples and sample groups can be compared largely independent of existing grain size differences [43], shows clearly increased enrichments of Cd, Mn, Ni, and Zn in SS and/or RS, which coincides with the grouping of these elements by the PCA. The preferred enrichment of these elements within aquatic sediments is explainable by additional inputs into the water systems via leaching processes from the soils of the catchment areas. This is evidenced by the comparatively high mobilizable fractions of the total element concentrations, the significant correlations to the pH value of the samples, and the joint grouping in the frame of the PCA. Further indications are given by the fact that Cd, Mn, Ni, and Zn are already increasingly dissolved under pH values of <5.5 [60,61]. Based on this process, these elements have a higher potential to be enriched in SS and RS, although the intensity of this enrichment can vary between single study areas (for details see Table S4). The CS, in turn, show the clearly lowest pH values in the comparison of the sample groups due to a stronger acid formation and leaching potential under humid climate, little acid-buffering rocks, and forest vegetation. It can be assumed that this element transfer occurring between a soil and a water body, if present, also includes anthropogenic substance accumulation in soils, as is known for the catchments of numerous reservoirs [62]. Similar to Pb, the concentrations of Cd and Zn are significantly correlated with TOC and $N_t$. It is likely that Cd and Zn are partially enriched by adsorption to organic matter or the formation of sulfides [63,64] in semi-terrestrial subsoils or organic-rich fine sediments after they enter surface water. It remains unclear how far the variable quantity and quality of organic substances between the different sampling groups—which can additionally impact metal binding [65]—might influence this process.

For Mn and Ni, the possibility of sulfide formation is assumed to be less significant, since Mn can retain a comparatively high mobility even under strongly reducing conditions and its hydroxides simultaneously represent a major adsorbent of Ni [66]. The influence of point and/or anthropogenic sources, where an increased release of Cd, Mn, Ni, and Zn could occur, can be largely excluded due to the selection of the water bodies studied (headwaters of mostly drinking water reservoirs).

## 5. Conclusions

The enrichments of As, Cd, Cr, Mn, Ni, Pb, V, and Zn confirm the sink function of reservoir sediments (RS) for potential contaminants. In addition, the comparison of RS, catchment soils (CS), and stream sediments (SS) indicates differences in the enrichment tendencies of individual metal(loid)s. Depending on a specific element, the enrichment can be influenced to varying degrees by grain sorting, the input of organic detritus, and a release from the soils by leaching processes. In particular, reservoirs impounding relatively

small water bodies with forested catchments are expected to be enriched by Cd, Mn, Ni, and Zn in RS and sediments of their inlets. On the one hand, the element release from CS is supposed to be particularly intensive in small water bodies due to a relatively high contact area between water and channel bed. On the other hand, the catchment areas of drinking water reservoirs usually have high proportions of forested areas, where under humid climatic conditions and poorly acid-buffering parent rocks a majority of acidified CS may enlarge the release of elements to running water. Impoundments of larger water bodies should be much less affected by this process. Investigations of the mobilizable element fractions in the field-fresh RS show that, with the exception of Mn, no relevant negative influence on the element concentrations of the water column can be assumed from the sediments. In the case of the investigated reservoirs, a deterioration in drinking water quality due to metal enrichments is not to be assumed as long as the bottom sediments remain undisturbed and covered by the water column. However, the apparently quite significant proportions of enriched metals that enter RS from CS due to leaching processes lead to the assumption that the enrichment of potential contaminants in RS will continue in the future, even if metal(loid) emissions from point sources are further reduced. Due to the increasing need for renovation resulting from the increasing age of reservoirs, the management of RS will be a future challenge, for which sufficient options of landfilling and recycling must be provided. For dam operators, synergy effects could arise if these options are developed together with other service providers involved in the management of dredged material, such as watercourse maintenance facilities.

**Supplementary Materials:** The following are available online at https://www.mdpi.com/article/10.3390/app12052277/s1, Table S1. Precautionary values according to the German Federal Soil Protection Ordinance (mg kg$^{-1}$). Table S2. Rotated component matrix of the three analyzed components. Table S3. Percentage of easily extractable trace metal concentrations on the aqua regia soluble concentrations of air-dried reservoir sediments (RS). Table S4. The element-specific enrichment factors (EF) of reservoirs with different catchment sizes.

**Author Contributions:** Conceptualization, J.H. and C.O.; methodology, J.H. and C.O; sample acquisition T.B., M.K., T.B., J.H., C.J.W., C.O.; chemical analyses J.H., T.B., M.K., A.M.; grain size analysis M.T.; element analyses, data validation, software, J.H.; writing—original draft preparation, J.H, C.O., C.J.W.; writing—review and editing, J.H, C.O., C.J.W. All authors have read and agreed to the published version of the manuscript.

**Funding:** We thank the Federal Ministry for the Environment, Nature Conservation, Nuclear Safety and Consumer Production for funding the costs necessary to publish this article.

**Institutional Review Board Statement:** Not applicable.

**Informed Consent Statement:** Not applicable.

**Data Availability Statement:** The data presented in this study are available on request from the corresponding author.

**Acknowledgments:** Our sincere thanks go to the operators of the investigated dams (Wasserverband Siegen-Wittgenstein; Wasserzweckverband Landkreis Birkenfeld, Landestalsperrenverwaltung Sachsen) for enabling and supporting the sampling campaigns. We also thank the Working Group for Geography at the University of Koblenz-Landau, the Department of Geography of the University of Marburg, and the Federal Institute of Hydrology for providing the field equipment.

**Conflicts of Interest:** The authors declare no conflict of interest.

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
