# Peer review of "Catchment Soil Properties Affect Metal(loid) Enrichment in Reservoir Sediments of German Low Mountain Regions"

_applsci, doi:10.3390/app12052277_

Round 1

Reviewer 1 Report

The manuscript goes on the comparison between concentrations and Enrichment Factors (EF) of metals in sediments collected in German reservoirs and their streams and in soils collected nearby the reservoirs. The final aim was to analyze soils as sources of metals for reservoirs, which are then accumulated in fine sediments. Sediment management is of critical importance in reservoirs, therefore the topic of the manuscript is interesting and of high relevance for publication in Applied Sciences. However, some points need to be better clarified. In particular:

- about comparisons between concentrations (Figure 2): concentrations of metals are analyzed on the < 2 mm grain size fraction, both in streams and in reservoirs. Moreover, values in some groups seem highly dispersed. Since coarser grain size are present in streams (Table 2) and metals are mainly accumulated in the finest fractions, the comparison should be carried out considering concentrations in the < 63 µm fraction, to reduce bias determined by differences in the granulometry. As an alternative, normalization on Al can be used.

- regarding EF calculation: the authors used for EF calculation the mean concentrations of elements in European river sediments derived from Foregs Atlas. First, I could not find in the Atlas the mean concentration for Al: did the author derive this value (5.4%) from another source? Second: by using the mean values in European stream sediment, the denominator of the equation is a constant for each element in all study area. Does this make sense, or a simple normalization of concentration to Al could be more realistic? For example, at lines 360-363, a local high geogenic background for As is mentioned, which is not considered in EF calculation.

- lines 259-280: statistical significance of differences between groups should be considered in this part of text, by rephrasing some sentences. For example, EF of V is not lower in RS than in the other groups (line 262) according to statistical analysis (Figure 3). From Figure 3, it seems that a few comparisons show significant differences.

- PCA analysis: what is the aim of this analysis, which considers concentrations in all samples (RS, SS and CS)? The groups of elements obtained along the three components are never mentioned in discussion. Only correlations of Table 4 are discussed. Therefore, the aim of the PCA should be clarified in paragraph “statistics” and the significance of the PCA results should be better clarified in discussion.

- I wonder if the results of the manuscript (i.e. that some metals derive from soils) would be confirmed by considering single case-studies (single reservoirs): maybe the authors may report results for some of the investigated reservoirs. Are differences between RS, CS and SS still confirmed, by considering a single case study?

Minor suggestions:

-Abstract, line 22: “from the gradation of sediment grains”: do the author mean “from accumulation of fine-grained sediments”? please, rephrase.

- Figure 2: statistical difference between Pb concentrations in CS vs RS should be controlled: values show strong overlapping. In capture, please add a description of box plot symbols: black square = mean, box = …., whiskers = …. Moreover, caption should report the statistical test used for comparisons and level of significance.

- Figure 3: In capture, please add a description of box plot symbols: black square = mean, box = …., whiskers = …. Moreover, caption should report the statistical test used for comparisons and level of significance.

- lines 297-298: correlation between Al and EF values can’t be performed, because Al is used for EF calculation itself.

- Table S1: please, “DS” should be corrected with “RS”.

- Conclusions: do metal concentrations in reservoir water confirm that soil leaching is a source of metals? Any (preliminary) data?

Author Response

Dear reviewer,

thank your very much for your constructive-critical and fair review.

You can find the answers to your single suggestions in the file attached.

Best regards, Jens Hahn.

Reviewer 2 Report

Dear authors, I believe that your article represents an adequate scientific contribution to the topic of study. I suggest the following modifications:

1.- I think the title of the manuscript, while very explanatory, should be shorter. Please synthesise the key information to generate a more concise and direct title that captures the readers' attention.

2.- Abstract: information that introduces the study needs to be incorporated, why is your research important? Likewise, I advise you to specify the meaning of the acronyms of the metals analysed.

3.- Line 42: please remove "cf".

4.- Line 48: specify in more detail what the legal levels are in terms of pollutant concentration, this is highly important information.

5.- Figure 1: please add the scale of the figure and its orientation to know which way is north.

6.- 2.2 Soil and sediment sampling: this sub-section needs to incorporate references to support the methodology used. I understand that this is the procedure that the authors have used to collect the samples, but often the methodologies used are similar to those used in other studies.

7.- 2.3 Laboratory work: I suggest changing the title of this sub-section to "sample processing". As in the previous comment, such processing should be supported by references.

8.- Lines 197-211: I believe this information does not belong in this sub-section. Please review and correct.

9.- 3.1. Sample characterization: the information in this sub-section is repetitive with the data in the Table. I suggest that the authors synthesise the information and discuss only the most relevant aspects, without repeating the data in the table. Perhaps, if they want to make comparisons, they can calculate and talk about increases or decreases of the observed variables.

10.- Table 4: in the table footnote they talk about p-values, but later in the table I think the values correspond to Spearman's statistic. Please clarify, it is confusing as it is now.

Author Response

Dear reviewer,

thank you very much for your constructive-critical and fair review.

You can find the answers to your single comments in the file attached.

Best regards,

Jens Hahn.

Round 2

Reviewer 1 Report

The authors provided detailed answers to all comments/suggestions and integrated in the original version of the manuscript the requested details and modifications. I think that the manuscript is now clear and well conceived and it can be published in the present form.